# Measuring women's empowerment in aquaculture in northwestern Bangladesh using a project level women's empowerment in fisheries index (pro-WEFI)

**Rahma I. Adam**[1]*, **Surendran Rajaratnam**[2], **Farha Deba Sufian**[3], **Lucy Njogu**[1]

**1** WorldFish Kenya, C/O International Livestock Researtablech Institute, Nairobi, Kenya, **2** Center for Research in Psychology and Human Well-Being, Faculty of Social Sciences & Humanities (FSSK), Universiti Kebangsaan Malaysia, Bangi Selangor Dahul Ehsan, Malaysia, **3** 17 Henry Macaulay Avenue, Kingston Upon Thames, United Kingdom

* R.Adam@cgiar.org

**Data Availability Statement:** All relevant data are within the manuscript and its Supporting Information files.

## Abstract

Gender equality and women's empowerment have been increasingly emphasised in food production systems, including fisheries and aquaculture. Accurate assessment and understanding of the state, progress and changes in women's empowerment in the sub-sectors is required. We applied the project level Women's Empowerment in Fisheries and Aquaculture Index (pro-WEFI), which is based on the project-level women's empowerment in agriculture index (pro-WEAI) to standardize the measurement of women's agency and empowerment in fisheries and aquaculture. Drawing on a survey conducted in north-western Bangladesh, we examined quantitative pro-WEFI data collected from 217 households engaged in aquaculture. Only 33% of the women and 48% of the men in the sample achieved empowerment in aquaculture, attaining scores of 0.75 and above. The mean disempowerment score (1-3DE) revealed that both women and men failed to achieve adequacy on average in nearly 28% of the indicators. Nearly 40% of the dual adult households did not attain gender parity with women achieving lower adequacy scores than men from the same household. Women's disempowerment was primarily driven by lack of autonomy in their use of income (18.5%), inability to visit important locations (17.4%), and inadequate access to and decision making on financial services (13.4%). Our findings emphasize the significance of conducting comprehensive assessments of women's empowerment in aquaculture initiatives and its various domains and indicators inform the development of targeted and effective interventions. By identifying domains where gender inequality is most pronounced, projects can better design interventions to create targeted impacts in critical areas.

## 1. Introduction

Although gender equality and women's empowerment have been increasingly emphasised (and their importance gradually realised) in the agendas of development actors, including

**Funding:** Bill and Melinda Gates Foundation CGIAR Gender Impact platform The funders had no role in study design, data collection and analysis, decision to publish, or preparation of the manuscript.

**Competing interests:** The authors have declared that no competing interests exist.

governments, their conceptual and political significance remains inadequate [1]. A lack of standardised and reliable measurement tools poses barriers for development actors to accurately assess and understand the progress and impact of their efforts towards reducing inequalities and progressing towards women's empowerment, especially with respect to marginalised and disadvantaged groups [2]. However, the use of context-specific measures of empowerment facilitates the identification of dimensions of empowerment that are pertinent to the local context [3]. These identified dimensions can serve as a basis for the development of locally-relevant interventions and policies that effectively target and address the specific needs and challenges faced by disadvantaged groups. The development and validation of context-specific measurement tools are important mechanisms to advancing women's empowerment and efforts to reduce inequalities.

Measures of women's empowerment can be traced back to a typology of power rooted in the works of Freire [4] on freedom and Lukes [5] on power [6]. Rowlands [7] further defined the concept of empowerment against the backdrop of gender and development. The concept can be used in a variety of ways, depending on the specific intention of its users [7, 8]. This variety can be seen in ways that empowerment has been used in the development of interventions with women [7].

The Women's Empowerment in Agriculture Index (WEAI) builds upon Kabeer's definition of empowerment which is "about the process by which those who have been denied the ability to make strategic life choices acquire such an ability" [8, p. 435]. The WEAI was published in 2012 by the International Food Policy Research Institute (IFPRI), Oxford Poverty and Human Development Initiative (OPHI), and USAID's Feed the Future. It is intended to be a comprehensive and standardised measure for directly assessing the level of empowerment, agency, and inclusion of women in the agricultural sector [9]. The WEAI was initially designed as a standardised means to assess changes in women's empowerment that could potentially result from the implementation of the United States government's Feed the Future Initiative (FTF) across countries, regions, and population subgroups [6, 10]. To date most applications of the WEAI measure women's empowerment in the agricultural sector and may not adequately capture the unique dynamics and challenges that are specific and inherent to other sectors such as livestock and aquaculture [11].

Women in the aquaculture and fisheries sectors face various unique challenges. These might include the time burden of engaging in shrimp farms [12], for instance, or little control over the usage or outcomes of their own household ponds [13]. However, the generalised lack of sex-disaggregated and gender data from aquaculture and fisheries interventions [14, 15] hampers the development of an understanding of the specific needs, constraints and opportunities of women in the sector. Good quality sex-disaggregated and gender data are needed to inform research and development in the sector, especially for women and other marginalised groups [16].

Although various tools and approaches have been used over the years to suit specific types of fisheries and aquaculture sectors across various countries to measure women's empowerment [17–20] these approaches largely lacked standardization and/ or validation. In this paper then we use the project-based Women's Empowerment in Fisheries and Aquaculture Index (pro-WEFI), which was derived from the project level WEAI and WEAI respectively to assess on the status of and barriers in women's empowerment in north-western Bangladesh aquaculture sector. Several versions of pro-WEFI have been developed and used in studies done in Bangladesh [16, 20, 21], India and Zambia [19, 20, 22], thus the pro-WEFI we used, was developed based on the findings of those studies. Most importantly, the major reasons for deploying this tool are:

(*i*) to understand the level of gender parity in the households under study; and (*ii*) to able to decipher exactly where women have challenges as they engage in the aquaculture sector (culture fish farmers). The understanding gathered from this study will aid researchers and development practitioners to design interventions that fit for purpose, hence advancing the process of addressing gender inequities matters in the aquaculture sector. We employ robustness tests and checks to validate the selection of indicators of the pro-WEFI and the weighting structure of the index. We compare our key findings with other projects in Bangladesh that assess women's empowerment in agriculture. This tool has enabled us to get the following information: (*i*) the level of gender parity in the households under study; and (*ii*) able to decipher exactly where women have challenges as they engage in the aquaculture sector. The understanding gathered from this study will aid researchers and development practitioners to design interventions that fit for purpose, hence advancing the process of addressing gender inequities matters in the aquaculture sector (culture fish farmers).

The rest of the paper is structured as follows: section 2 describes the methodology on data collection and sampling, as well as cognitive testing that was carried out before the actual data collection, and the framework used for data analysis. Section 3 presents the findings which cover demographic characteristics of respondents and women's empowerment level compared with men in the aquaculture sector. Findings are followed by discussion, where we go into detail on what our findings mean, and lessons learned from the study. We conclude the paper, by having a conclusions section.

## 2. Methodology

The Pro-WEFI instrument was implemented through the Increasing Income, Diversifying Diets, and Empowering Women in Bangladesh and Nigeria (IDEA) project in Bangladesh. The IDEA project was carried out in Rajshahi and Rangpur divisions (northwestern Bangladesh), with the purpose of enhancing the incomes, diets and nutrition of smallholder aquaculture involved families, as well as to increase opportunities for women's empowerment through aquaculture initiatives. Informed consent was sought from the participants of the study before administering the questionnaire. The authors of this manuscript do not have access to information that could identify individual participants during or after the data collection. Approval for the study was granted by the Ethics Committee of the University of Dhaka, Institute of Health Economics, and the study was performed in line with the principles of the Declaration of Helsinki.

### 2.1. Data collection

The study was conducted in two phases. The first phase involved a cognitive testing of the questionnaire and the second phase involved implementation of the pro-WEFI questionnaire.

**2.1.1. Cognitive testing.**   The pro-WEFI was cognitively tested in Bengali-speaking communities living in West Bengal, India in September 2021. The high COVID-19 situation and the need to get travelling documents for the researchers to travel to Bangladesh to carry out the research, prevented us from conducting the cognitive study directly in Bangladesh. Testing entailed administering the pro-WEFI survey questions whilst obtaining additional responses on the survey to improve the quality of responses and obtain the intended information. This involved interviewing survey respondents to explore four stages of the cognitive process when answering the survey questions: 1) comprehension, 2) retrieval, 3) judgement, and 4) response. Cognitive testing was conducted with 52 respondents including 14 men and 38 women from dual headed (N = 48) and female headed (N = 4) households. The respondents and communities were chosen to be representative of a variety of aquaculture contexts and thus fieldwork

was conducted in three agro-climatic zones. All respondents were interviewed alone (i.e., without the presence of other family members or neighbours). The cognitive testing of the pro-WEFI tool only used the structured household questionnaire survey, deployed for a man and a woman of the household (husband and wife) who practices aquaculture (culture fish farming).

Out of the 12 modules in the Pro-WEFI, presented in this paper, eleven were cognitively tested. The time use module was not tested because it has been extensively tested and revised as part of WEAI cognitive testing and piloting, including in Bangladesh. Upon completion of cognitive testing, each tool was revised by removing or modifying questions with high levels of cognitive difficulties. The results informed, which versions of certain modules and/or questions to use as well as how to phrase questions better for the respondents to understand. The revised tool was then validated in Bangladesh in 2022. Further information on cognitive testing can be found in McDougall et al. [23].

**2.1.2. Pro-WEFI implementation.** The study employed a structured pro-WEFI household survey. In terms of sampling, the study was conducted from July through September of 2022 in the Bogura, Naogaon, and Rangpur divisions of Bangladesh (See S1 Datasets). Two hundred and seventeen (217) women and 208 men were interviewed from a total of 217 households in 30 villages engaged in fish farming activities. The respondents interviewed as part of the pro-WEFI implementation study included men and women beneficiaries of the IDEA project's nutrition, aquaculture, and gender training.

**Fig 1** provides an overview of the domains of the pro-WEFI.

## 2.2. Data analysis

The collected datasets were analysed using STATA software version 16. The analysis is based on the domains and indicators of pro-WEFI. Pro-WEFI quantifies women's empowerment status in the aquaculture and fisheries by assessing women's agency using the quantitative survey instrument adapted from the pro-WEAI questionnaire. For the definitions and adequacy cut-offs for every indicator that was applied to calculate the cognitively tested pro-WEFI index please refer to McDougall et al. [23]. Each indicator is made up of various questions that are given to the interviewed women and men that are included in the study. Finally, a weighted index is calculated using twelve indicators, each indicator received a weight of 1/12, following the pro-WEAI method of calculation.

The composition of pro-WEFI retains important properties from the pro-WEAI by assessing agency of women and men in the fisheries and/or aquaculture across three key domains: *(i)* intrinsic agency, which implies power within, *(ii)* instrumental agency, which can also be referred to as power to, and *(iii)* collective agency, which is recognized as power with [6]. Pro-WEFI methodology builds on all 12 indicators from pro-WEAI by adding and updating existing questions with fisheries and aquaculture specific scenarios and options.

Intrinsic agency consists of four indicators: self-efficacy, autonomy in income, respect among household members, and attitudes about intimate personal violence (IPV) against women. Pro-WEFI includes new scenarios and statements for assessing *self-efficacy* and *attitudes toward IPV against women* compared to pro-WEAI, with subsequent adjustments made to the adequacy cut-offs to be theoretically consistent. Such changes from pro-WEAI were made based on the outcomes of the cognitive test conducted prior to the survey.

The instrumental agency element in pro-WEFI is composed of six indicators: ownership of land, access to and decisions on financial assets, input in productive decisions, work-life balance, control over use of income, and visiting important locations. Pro-WEFI adds key decisions pertaining to fishing and post-fishing activities to the list of work activities used in pro-WEAI to assess respondent's *input in productive decisions* and *control over income*.

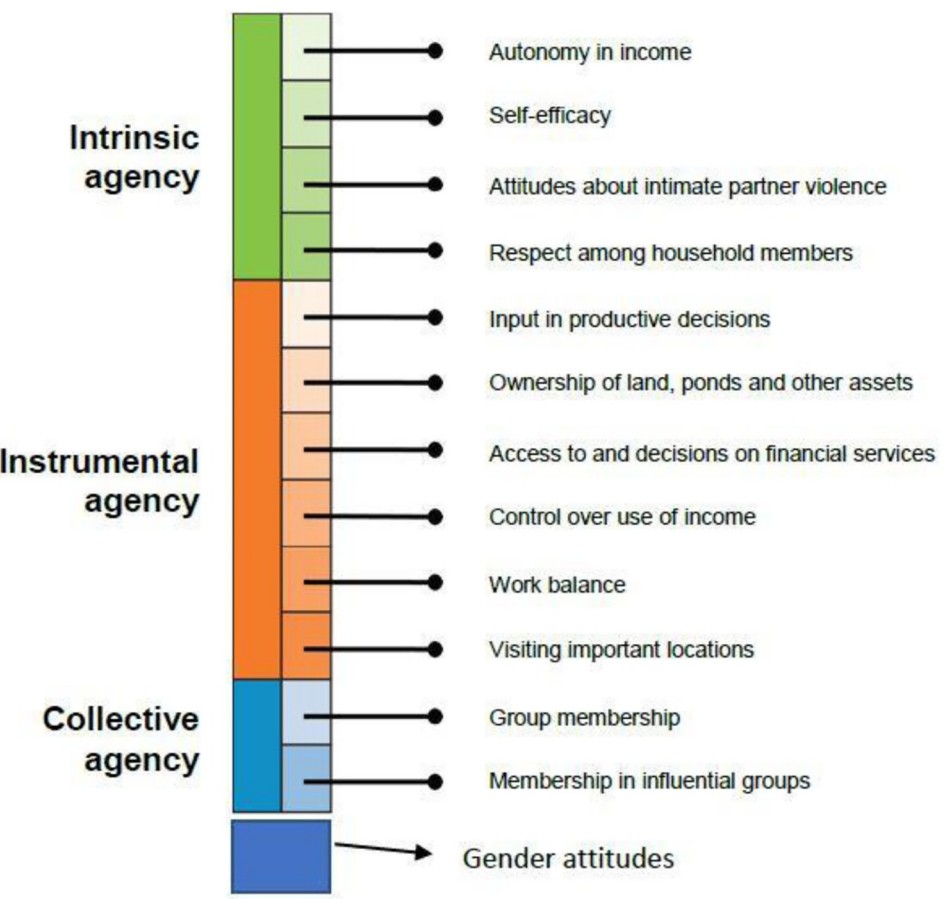

**Fig 1. Domains and indicators of pro-WEFI.** Source: [23].

Subsequently, the *ownership of land and assets* indicator were revised to account for ownership of ponds and access to gleaning area. These are essential resources to households engaged in fisheries and/or aquaculture. Finally, collective agency is composed of two indicators assessing membership in influential groups, and group membership. Pro-WEFI expands on the list of groups questioned in the pro-WEAI by adding groups that households who participate in fishing and post fishing activities are likely to associate with. Such modifications allow pro-WEFI methodology to track and monitor the various dimensions of empowerment of women participating in the aquaculture and fisheries sector and diagnose the impediments to their agency.

*Index construction*: responses from the pro-WEFI survey are then used to categorize men and women as adequate or inadequate for all indicators using the corresponding thresholds exhibited in McDougall et al., [23]. Thus, 12 binary variables are deduced, with 0 representing inadequacy in the indicator and 1 indicating adequacy. A weighted average of all indicators, using equal weights of 1/12, is then computed to compute individual empowerment scores for both women and men sampled. Following pro-WEAI, respondents scoring 0.75 or higher are classified as empowered, and conversely, as disempowered if scores below 0.75, i.e., found inadequate in 4 or more out of the 12 pro-WEFI indicators.

Pro-WEFI, following the pro-WEAI methodology, is measured as a weighted mean of two sub-indices. They are the three Domains of Empowerment Index (3DE) and Gender Parity Index (GPI), where the former receives a 90% weight, and the latter is assigned the remaining

10%. The *3DE score* is the aggregated individual scores, which measures the respondents' empowerment across three domains. It ranges from 0 to 1, with higher scores representing greater empowerment. GPI is computed using individual scores of the man and woman from the same household to account for the extent of gender equality in the final pro-WEFI index. More information on how the individual indicators are joined together to calculate the 3DE, GPI and composite index can be found in Malapit et al., [6].

## 3. Results

### 3.1. Demographic characteristics

Summary statistics of the population sample that participated in the study are presented in **Table 1**. As noted above, the pro-WEFI was administered in 217 households in Bogura, Naogaon, and Rangpur districts of northwestern Bangladesh. All households surveyed were dual adult households, composed of at least one male and one female above eighteen years of age. Sixty-three percent (63.1%) of the women surveyed were aged between 26 and 45 years, and were relatively younger than the male respondents. Less than 10% of both men and women had never attended school. For those who completed primary school, 45.2% were women and 35.6% were men. Finally, most men (96.2%) and women (98.6%) in the sample reported being married and identified as Muslims (88% and 88.5% respectively).

### 3.2. Aggregate pro-WEFI scores

**Table 2** summarizes key results from the aggregate indices. The aggregate pro-WEFI score for women is 0.73, which is a weighted mean of the women's 3DE score of 0.72 and the GPI score of 0.90. Thirty-three percent (33%) of the women and 48% of the men attained scores of 0.75 and above, classified as achieving empowerment. The intensity of disempowerment for both

**Table 1. Demographic characteristics of the respondents by sex.**

| Variable | Percent of respondents | |
|---|---|---|
| | **Women** | **Men** |
| *Age group* | | |
| 16–25 | 9.2 | 4.3 |
| 26–45 | 63.1 | 47.1 |
| 46–65 | 27.2 | 45.7 |
| >65 | 0.5 | 2.9 |
| *Education* | | |
| Never attended school | 6 | 7.7 |
| Below Primary | 17.1 | 11.1 |
| Primary | 45.2 | 35.6 |
| Secondary | 16.6 | 22.1 |
| Higher secondary or above | 15.2 | 23.6 |
| *Marital Status* | | |
| Never married | 0.5 | 3.8 |
| Married | 98.6 | 96.2 |
| Divorced/Separated/Widowed | 0.9 | 0 |
| *Religion* | | |
| Islam | 88.5 | 88 |
| Hinduism | 11.5 | 12 |
| Dual Adult Household | 100 | 100 |

**Table 2. Pro-WEFI scores, by sex of respondent.**

| Indicator | Women | Men |
|---|---|---|
| Number of observations | 217 | 208 |
| **3DE score** | **0.72** | **0.79** |
| Disempowerment score (1 – 3DE) | 0.28 | 0.21 |
| % achieving empowerment | 33% | 48% |
| % not achieving empowerment | 67% | 52% |
| Mean 3DE score for not yet empowered | 0.57 | 0.60 |
| Mean disempowerment score (1 – 3DE) | 0.43 | 0.40 |
| **Gender Parity Index (GPI)** | **0.90** | |
| Number of dual-response households | 208 | |
| % achieving gender parity | 60% | |
| % not achieving gender parity | 40% | |
| Average empowerment gap | 0.24 | |
| **Pro-WEFI score** | **0.73** | |

women and men are reflected by the mean disempowerment score (1-3DE), which suggests that men and women who had scores below 0.75 failed to achieve adequacy on average in nearly 28% of the indicators. Sixty percent (60%) of women achieved gender parity; the remaining 40% of the women had lower empowerment scores on average than the men surveyed from the same households. The average percentage shortfall in empowerment scores for those women who do not have gender parity was 0.24 relative to the men.

To evaluate the validity of the scores obtained in the pro-WEFI pilot study presented in Table 2, we compared these findings with results from other projects in Bangladesh that assessed women's empowerment in the productive sector, particularly in agriculture. Specifically, we analyzed the Pro-WEFI results for women in intervention villages of the ECOFISH II project, the WEAI scores from the baseline study of the Bangladesh Feed the Future (FTF) Survey 2011, and the pro-WEAI scores from both control and treatment groups in the Food and Agricultural Approaches to Reducing Malnutrition (FAARM) trial, as shown in Table 3.

The proportion of women achieving empowerment varied significantly across these studies, ranging from just 4 percent in the FAARM trial to 33 percent in our study. Notably, the percentage of women who achieved empowerment in our study (33 percent) is more than double that observed in the Pro-WEFI analysis of women in the ECOFISH II project (14 percent). Additionally, gender parity was achieved in 60 percent of dual-adult households in our study, compared to only 31 percent in the ECOFISH II project.

Furthermore, the 3DE, GPI, and final pro-WEFI scores from our study are slightly higher than those reported in the WEAI and pro-WEAI results from the three projects considered. These scores are closely comparable to the pro-WEAI results for women in the treatment areas of the FAARM trial.

## 3.3. Uncensored and censored headcount ratios

While the aggregate pro-WEFI index, 3DE, and GPI scores provide valuable measures of empowerment at the project level for both genders, combining these high-level indexes with sub-indicators and subcomponents offers more comprehensive insights for tailoring interventions to advance gender equality. Therefore, the decomposability of the pro-WEFI into dimensions and indicators allows for identifying drivers of change, areas for improvement, and gendered differences in adequacy achievements across various types of agency and indicators. Table 4 shows results of the uncensored and censored headcount ratios of inadequacy for the

**Table 3. Comparison of pro-WEFI results to other women's empowerment indices from Bangladesh.**

| Indicator | Pro-WEFI (12 Indicators) | Pro-WEFI ECOFISH II [1] (12 indicators) | WEAI FTF [2] (10 indicators) | Pro-WEAI FAARM trial [3] (control/ treatment) (12 indicators) |
|---|---|---|---|---|
| Number of observations | 217 | 364 | 1938 | 227 / 230 |
| **3DE score** | **0.72** | **0.57** | **0.65** | **0.50 / 0.66** |
| % Achieving empowerment | 33% | 14% | 25% | 4% / 24% |
| Mean 3DE score for not yet empowered | 0.57 | 0.49 | 0.47 | 0.47 / 0.56 |
| **Gender Parity Index (GPI)** | **0.90** | **0.79** | **0.80** | **0.76 / 0.88** |
| Number of dual-adult households | 208 | 380 | 1657 | 205 / 215 |
| % Achieving gender parity | 60% | 31% | 39% | 29% / 54% |
| Average empowerment gap | 0.24 | 0.31 | 0.33 | 0.34 / 0.26 |
| **Final Index Score** | **0.73** | **0.59** | **0.66** | **0.52 / 0.69** |

[1] https://journals.sagepub.com/doi/epub/10.1177/21582440241250114

[2] Bangladesh Feed the Future Survey 2011.

[3] https://www.sciencedirect.com/science/article/pii/S0305750X22001917

12 pro-WEFI indicators by gender. Uncensored headcount ratios reflect the percent of respondents found inadequate in a given indicator, irrespective of their overall empowerment status. Whereas the censored headcount reflects the percent of interviewee who are classified as disempowered, and also are inadequate in a given indicator, as defined by Alkire and Foster [9]. Since nearly two-thirds of the women and half the men in the sample are disempowered, the figures in censored and uncensored headcounts reflect differences.

Table 4 shows results of the uncensored and censored headcount ratios of inadequacy for the 12 pro-WEFI indicators by gender. The majority of women in the study revealed significant shortcomings in attaining autonomy in income (94%) and in their ability to access important locations (79%). Despite 63% of women claiming control over their income, their lack of autonomy in income implies that decisions regarding income allocation did not reflect their

**Table 4. Headcount ratios of inadequacy in pro-WEFI indicators.**

| | Uncensored headcount (ratio%) | | Censored headcount (ratio%) | |
|---|---|---|---|---|
| | Women | Men | Women | Men |
| *Intrinsic agency* | | | | |
| Autonomy in income | 94.0 | 76.0 | 63.1 | 40.4 |
| Self-efficacy | 38.2 | 11.5 | 30.9 | 8.7 |
| Attitudes about intimate partner violence against women | 2.3 | 2.9 | 2.3 | 2.4 |
| Respect among household members | 16.6 | 36.5 | 14.3 | 33.2 |
| *Instrumental agency* | | | | |
| Input in productive decisions | 5.5 | 2.9 | 5.5 | 2.4 |
| Ownership of land and other assets | 1.4 | 0.5 | 1.4 | 0.5 |
| Access to and decisions on financial services | 56.2 | 34.1 | 45.6 | 24.5 |
| Control over use of income | 38.7 | 31.3 | 26.3 | 21.6 |
| Work balance | 7.8 | 25.0 | 6.5 | 16.3 |
| Ability to visit important locations | 78.8 | 8.2 | 59.4 | 4.3 |
| *Collective agency* | | | | |
| Group membership | 42.9 | 58.2 | 41.0 | 47.1 |
| Membership in influential groups | 48.4 | 72.6 | 44.7 | 50.5 |

own values. Moreover, a notable proportion of women (56%) lacked access to or authority over credit and financial services. Women, however, reported substantial involvement in decision-making regarding aquaculture production. Ownership of assets and land exhibited high adequacy scores, as did achieving a work-life balance, contrary to expectations from prior research indicating challenges faced by rural women in achieving this equilibrium.

Comparing the censored and uncensored headcount ratios is crucial for understanding the limitations of indicators concerning empowerment status. When the uncensored and censored headcounts are closely aligned, it indicates that the inadequacy primarily stems from disempowered individuals. This scenario is observed in the censored and uncensored headcounts for women's achievement of group membership, membership in influential groups, self-efficacy, access to credit, and restricted mobility. It suggests that most women who are inadequate in these indicators also struggle to achieve empowerment overall.

## 3.4. Distribution of inadequacies by sex

Fig 2 compares the distribution of the number of inadequate indicators across genders. Respondents with 4 or more inadequate indicators out of the 12 (i.e. with an individual score below 0.75 out of 1) are classified as disempowered, represented by the shaded area in Fig 2. Overall, men have fewer inadequacies than women. This indicates that, on average, women experience a higher frequency or severity of inadequacies compared to men. However, the intensity of disempowerment experienced by men and women in the sample are very comparable, as both distributions in the figure follow a very similar pattern for 4 or more inadequacies. This observation is line with the results exhibited in the high-level indices summary in Table 2, where both headcount of respondents not achieving empowerment and the average disempowerment score for women are nearly the same as that of men, suggesting similar intensity of disempowerment.

## 3.5. Key sources of disempowerment

One of the key objectives of developing pro-WEFI is to identify the key obstacles facing improvements in the empowerment status and agency of women who engage in fisheries or

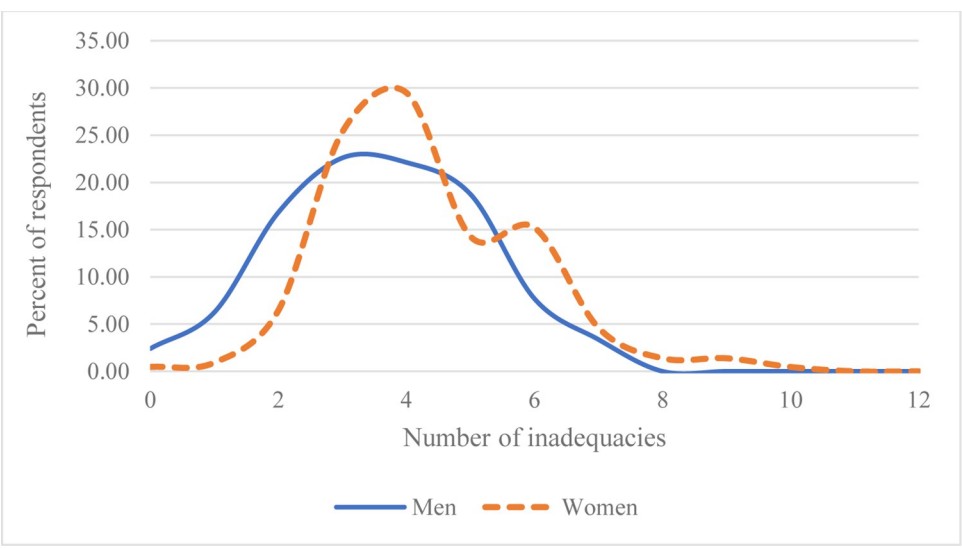

**Fig 2. Distribution of inadequacies.** Notes: Shaded box indicates disempowered respondents, i.e., those who are inadequate in four or more indicators. *Source*: *Analysis by the authors based on the* structured pro-WEFI household survey *in Bangladesh (2022)*.

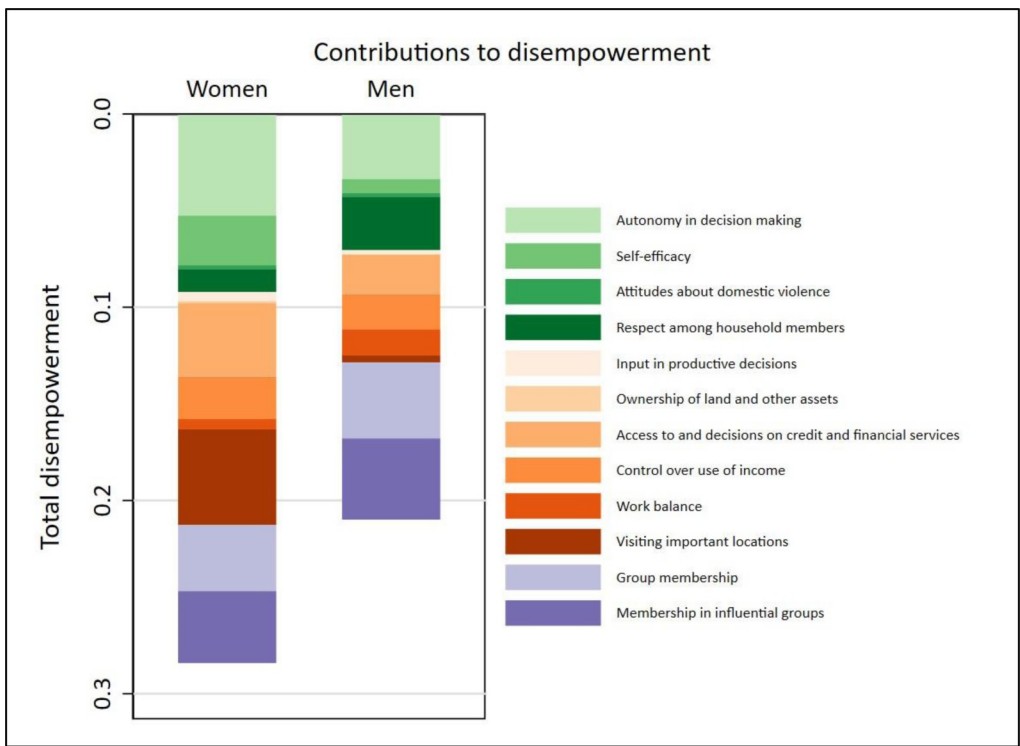

**Fig 3. Contributions of each pro-WEFI indicator to disempowerment by sex of the respondents.** *Source*: *Analysis by the authors based on the* structured pro-WEFI household survey *in Bangladesh (2022).*

aquaculture. The diagnostics presented in Fig 3 and Table 5 provide key insights in highlighting the highest contributors to the disempowerment of the women in the sample. Fig 3 displays the absolute contribution of each of the 12 pro-WEFI indicators, represented by the different coloured bars, to the disempowerment score (1-3DE) of the sampled women and men. The total disempowerment scores, 0.28 for women and 0.20 for men, are reflected in the depth of each bar in the figure.

Table 5 exhibits the proportional contribution of each indicator to the disempowerment of men and women respondents who have not yet achieved empowerment. (For ways of computing, see [6]. Such a diagnostic helps identify the key challenges facing the disempowered in the sample. For women, the biggest impediments to their achievement of empowerment comes from their lack of autonomy in the use of income (18.5%), inability to visit important locations (17.4%), their lack of access to and decision making on financial services (13.4%), and inadequate self-efficacy (9.1%). We also find membership in influential groups (20%) and group membership (18.7%) making the highest contributions to the disempowerment of men in the population. Additionally, respect among household members (13.2%) poses a threat to men's achievement of empowerment.

## 3.6. Intrahousehold patterns of empowerment

The sample from the pilot study of pro-WEFI comes from all dual adult households, where both women and men from the same household self-report their adequacies or inadequacies across several indicators. This allows us to examine the intrahousehold patterns of empowerment by comparing their individual weighted average scores, using the pro-WEFI quantitative methodology discussed earlier. Among the 208 households with responses from both women

**Table 5. Relative contributions of each indicator to disempowerment.**

|  | Proportional contribution to disempowerment (%) | |
|---|---|---|
|  | **Women** | **Men** |
| *Intrinsic agency* |  |  |
| Autonomy in income | 18.5 | 16.0 |
| Self-efficacy | 9.1 | 3.4 |
| Attitudes about intimate partner violence against women | 0.7 | 1.0 |
| Respect among household members | 4.2 | 13.2 |
| *Instrumental agency* |  |  |
| Input in productive decisions | 1.6 | 1.0 |
| Ownership of land and other assets | 0.4 | 0.2 |
| Access to and decisions on financial services | 13.4 | 9.7 |
| Control over use of income | 7.7 | 8.6 |
| Work balance | 1.9 | 6.5 |
| Ability to visit important locations | 17.4 | 1.7 |
| *Collective agency* |  |  |
| Group membership | 12.0 | 18.7 |
| Membership in influential groups | 13.1 | 20.0 |

and men, we find that 47.8% of the sampled households had men being adequate in more indicators than women (**Table 6**). Nearly one quarter of the surveyed households (24%) had the woman and the man adequate in equal number of indicators, while 28.2% had women adequate in more indicators than men. While 30.3% of the households had only men empowered, 16.3% of the households had only women empowered. We also find that only in 17.8% of the overall sample, both men and women are empowered, and in 35.6% of the households, neither the woman nor the man achieved empowerment.

## 3.7. Robustness

**3.7.1. Association between pro-WEFI indicators.** As discussed earlier, each indicator in the pro-WEAI, and therefore pro-WEFI, methodology is assigned an equal weight of 1/12. It is therefore a useful exercise to assess the pairwise associations between each indicator. We use Cramer's V to identify the strength of the correlation, where two variables are weakly correlated if Cramer's V is less than 0.3. Most pro-WEFI indicators are weakly correlated with each other, with one key exception (Table 7). The results show a strong correlation between group membership and membership in influential groups (Cramer's V = 0.82).

**Table 6. Intrahousehold patterns of empowerment.**

|  | % of dual-adult households |
|---|---|
| Male adequacy score > female adequacy score | 47.8 |
| Female adequacy score > male adequacy score | 28.2 |
| Female adequacy score = male adequacy score | 24.0 |
| Only male is empowered | 30.3 |
| Only female is empowered | 16.3 |
| Both male and female are empowered | 17.8 |
| Neither male nor female are empowered | 35.6 |

**Table 7. Association (Cramer's V) between pro-WEFI indicators.**

|  | Autonomy in income | Self-efficacy | Attitudes about domestic violence | Respect among HH members | Input in productive decisions | Ownership of land and other assets |
|---|---|---|---|---|---|---|
| *Intrinsic agency* |  |  |  |  |  |  |
| Autonomy in income | 1.000 |  |  |  |  |  |
| Self-efficacy | 0.089 | 1.000 |  |  |  |  |
| Attitudes about intimate partner violence against women | 0.068 | 0.042 | 1.000 |  |  |  |
| Respect among household members | 0.129 | 0.108 | 0.104 | 1.000 |  |  |
| *Instrumental agency* |  |  |  |  |  |  |
| Input in productive decisions | 0.055 | 0.040 | -0.034 | -0.020 | 1.000 |  |
| Ownership of land and other assets | 0.041 | 0.000 | -0.016 | -0.003 | -0.020 | 1.000 |
| Access to and decisions on financial services | 0.021 | 0.081 | 0.030 | -0.052 | 0.019 | 0.009 |
| Control over use of income | -0.151 | -0.165 | 0.036 | -0.115 | 0.262 | -0.021 |
| Work balance | -0.104 | -0.050 | -0.032 | -0.003 | -0.029 | -0.043 |
| Ability to visit important locations | 0.238 | 0.302 | 0.004 | -0.103 | 0.071 | 0.060 |
| *Collective agency* |  |  |  |  |  |  |
| Group membership | -0.308 | 0.088 | 0.014 | 0.348 | 0.162 | 0.048 |
| Membership in influential groups | -0.231 | 0.061 | -0.019 | 0.268 | 0.123 | 0.079 |
|  | Access to and decisions on financial services | Control over use of income | Work balance | Visiting important locations | Group membership | Membership in influential groups |
| *Instrumental agency* |  |  |  |  |  |  |
| Access to and decisions on financial services | 1.000 |  |  |  |  |  |
| Control over use of income | 0.172 | 1.000 |  |  |  |  |
| Work balance | -0.132 | 0.011 | 1.000 |  |  |  |
| Ability to visit important locations | 0.130 | -0.108 | -0.264 | 1.000 |  |  |
| *Collective agency* |  |  |  |  |  |  |
| Group membership | 0.036 | -0.050 | 0.054 | -0.054 | 1.000 |  |
| Membership in influential groups | 0.036 | -0.149 | 0.084 | -0.119 | 0.818 | 1.000 |

### 3.7.2. Redundancy between pro-WEFI indicators

To investigate how inadequacies are clustered, we further examine the redundancy among pro-WEFI indicators. Table 8 displays the redundancy between the 12 indicators, calculated using the definition of redundancy by Alkire et al. [24]. We observe significant redundancy among all pro-WEFI indicators, a finding consistent with Malapit et al. [6] regarding pro-WEAI indicators. Despite weak associations among indicators (Table 7), the high redundancy does not significantly affect the robustness of the weighted pro-WEFI index.

### 3.8. Sensitivity analysis

Next, we observed how the aggregate 3DE scores and the percent of respondents achieving empowerment for men and women vary by considering [1] different empowerment cut-offs, and [2] an alternative weighing scheme, following Alkire et al., [9].

**Table 8. Redundancy between pro-WEFI indicators.**

|  | Autonomy in income | Self-efficacy | Attitudes about domestic violence | Respect among HH members | Input in productive decisions | Ownership of land and other assets |
|---|---|---|---|---|---|---|
| *Intrinsic agency* |  |  |  |  |  |  |
| Autonomy in income | 1.000 |  |  |  |  |  |
| Self-efficacy | 0.841 | 1.000 |  |  |  |  |
| Attitudes about intimate partner violence against women | 1.000 | 0.978 | 1.000 |  |  |  |
| Respect among household members | 0.873 | 0.776 | 0.984 | 1.000 |  |  |
| *Instrumental agency* |  |  |  |  |  |  |
| Input in productive decisions | 0.984 | 0.962 | 0.973 | 0.955 | 1.000 |  |
| Ownership of land and other assets | 1.000 | 0.991 | 0.990 | 0.990 | 0.990 | 1.000 |
| Access to and decisions on financial services | 0.571 | 0.780 | 0.978 | 0.716 | 0.961 | 0.991 |
| Control over use of income | 0.476 | 0.696 | 0.978 | 0.699 | 0.996 | 0.989 |
| Work balance | 0.746 | 0.827 | 0.972 | 0.837 | 0.955 | 0.989 |
| Ability to visit important locations | 0.841 | 0.865 | 0.975 | 0.696 | 0.970 | 0.996 |
| *Collective agency* |  |  |  |  |  |  |
| Group membership | 0.127 | 0.787 | 0.976 | 0.891 | 0.991 | 0.995 |
| Membership in influential groups | 0.127 | 0.781 | 0.970 | 0.882 | 0.988 | 1.000 |

|  | Access to and decisions on financial services | Control over use of income | Work balance | Visiting important locations | Group membership | Membership in influential groups |
|---|---|---|---|---|---|---|
| *Instrumental agency* |  |  |  |  |  |  |
| Access to and decisions on financial services | 1.000 |  |  |  |  |  |
| Control over use of income | 0.724 | 1.000 |  |  |  |  |
| Work balance | 0.793 | 0.841 | 1.000 |  |  |  |
| Ability to visit important locations | 0.616 | 0.603 | 0.751 | 1.000 |  |  |
| *Collective agency* |  |  |  |  |  |  |
| Group membership | 0.564 | 0.626 | 0.858 | 0.531 | 1.000 |  |
| Membership in influential groups | 0.568 | 0.562 | 0.876 | 0.485 | 1.000 | 1.000 |

Table 9 exhibits the percent of men and women achieving empowerment across a range of empowerment cut-offs. Each column represents a different cut-off, i.e. adequacy in number of indicators required, to be classified as achieving empowerment. 0.75 cut-off used for pro-WEFI methodology, following pro-WEAI, represents achieving adequacy in 9 out of 12

**Table 9. Rank comparison of headcount ratios by gender for different empowerment cut-offs.**

|  |  | Number of indicators required to be empowered | | | | | | | | | | | |
|---|---|---|---|---|---|---|---|---|---|---|---|---|---|
|  |  | *12* | *11* | *10* | *9* | *8* | *7* | *6* | *5* | *4* | *3* | *2* | *1* |
| Male | Empowerment Headcount ratio | 2% | 2% | 9% | 48% | 48% | 70% | 97% | 97% | 100% | 100% | 100% | 100% |
|  | Rank | 1 | 1 | 1 | 1 | 1 | 1 | 1 | 1 | 1 | 1 | 1 | 1 |
| Female | Empowerment Headcount ratio | 0% | 0% | 1% | 33% | 33% | 63% | 92% | 92% | 97% | 100% | 100% | 100% |
|  | Rank | 2 | 2 | 2 | 2 | 2 | 2 | 2 | 2 | 2 | 2 | 2 | 2 |

**Table 10. Rank of 3DE score by gender for different weighing schemes.**

| | Equally weighted by indicator | | | Equally weighted by domain | | |
|---|---|---|---|---|---|---|
| | 3DE score | % achieving empowerment | Rank | 3DE score | % achieving empowerment | Rank |
| Women | 0.78 | 33 | 2 | 0.76 | 45 | 1 |
| Men | 0.80 | 48 | 1 | 0.69 | 30 | 2 |

indicators. The results suggest that across all cut-offs considered, a higher percentage of men achieve empowerment compared to women, i.e., men rank higher than women. Thus, the rank robustness results in Table 9 provide no evidence for altering the empowerment cut-off for pro-WEFI from 0.75.

Next, we compare 3DE scores and headcount ratio of empowerment for men and women considering two different weighing schemes: [1] 3DE score computed using equal weights for all indicators, followed by pro-WEFI [2] 3DE score computed using equal weights for three domains. The second scheme assigns a weight of 1/3 for each of the three domains used in pro-WEFI (Table 1), namely intrinsic, instrumental, and collective agency, and the indicators within each domain receive equal weights. Thus, the indicators within [1] intrinsic agency receive a weight of 1/12 each, [2] instrumental agency receive a weight of 1/18, and [3] collective agency receive a weight of 1/6. Table 10 compares 3DE scores and headcount ratio for men and women, and assigns ranks accordingly. Under the existing pro-WEFI scheme, i.e., scheme 1, men rank higher than women with a higher 3DE score (0.80) and a higher percentage achieve empowerment (48%) compared to women, thereby securing rank 1. However, the ranking changes under the alternative scheme, where more women achieve empowerment (45%) with a higher aggregate 3DE score of 0.76 relative to men. The results change noticeable for two main reasons. First, the two largest contributors to women's disempowerment, i.e., access to and decisions on credit, and ability to visit important locations, receive much lower weights (1/18) under scheme 2, compared to scheme 1 (1/12). This increases the aggregate empowerment score of women, together with their headcount ratio for those achieving empowerment. Conversely, the biggest contributors to men's empowerment, namely group membership and membership in influential group, receive double the weights under scheme 2 (1/6) compared to scheme 1 (1/12), lowering the aggregate empowerment score and the headcount ratio for men. Therefore, under scheme 2 the difference between result for men and women grow higher, in favour of the latter.

## 4. Discussion

The current study used the pro-WEFI, an adaptation of the pro-WEAI, to measure women's empowerment in the fish production context. The 3DE score implies the achievement of women in the sample across the twelve indicators, and it integrates both the headcount ratio of women who are disempowered, and the intensity of their disempowerment [6].

Our findings reveal a relatively higher disempowerment and gender parity for women as compared to men. These findings can be compared to those from projects in Bangladesh that measured women's empowerment in agriculture, particularly the women's pro-WEFI scores from the ECOFISH II project, women's WEAI scores from the baseline study conducted for Bangladesh Feed the Future (FTF) Survey 2011 [25] and the pro-WEAI scores from the control and treatment area women in the FAARM trial [26]. While the samples differ, the comparative analysis allows us to understand the similarities and differences in our findings relative to the projects considered.

Intriguingly, although the 3DE, GPI and the final scores of the pro-WEFI are slightly higher than the pro-WEFI, WEAI and pro-WEAI results for women in the control group among the projects considered, they compare closely to the pro-WEAI results observed for the women in the treatment area in the FAARM trial. This is likely to be driven by two main reasons. First, the sample size of the treated group in the FAARM trial and our study are very similar, compared to the much larger size of the FTF study, in which the sample is representative of the FTF zone population. Second, there are likely to be systematic differences in the characteristics of the women in the Pro-WEFI and FAARM sample compared to the women in the FTF study sample. The former are comprised of women who either engage in aquaculture production or who have received treatment through nutrition-sensitive agricultural programs. Therefore, we are likely to observe higher 3DE, GPI and the final index scores for the sampled women for pro-WEFI and pro-WEAI compared to those considered for WEAI and pro-WEFI stemming from the potential selection bias and the limitation of the small sample utilized in this study.

Even though the aggregate pro-WEFI index, 3DE and GPI scores are all valuable ways to aggregate empowerment at the project level for both men and women, making inferences to these high-level indexes together with the sub-indicators, and subcomponents give more well-rounded insights to help customize interventions in advancing gender equality. Thus, the decomposability of the pro-WEFI into the dimensions and indicators allows the identification of the drivers of change, areas of improvement, and the gendered difference in the adequacy achievements across different types of agency and indicators. Uncensored headcount ratios reflect the percent of respondents found inadequate in a given indicator, irrespective of their overall empowerment status whereas the censored headcount reflects the percent of the population who are classified as disempowered, and also are inadequate in a given indicator [9].

Since nearly two-thirds of the women and half the men are disempowered, according to our findings, the figures in censored and uncensored headcounts reflect disparities. Our study found a possible discrepancy between women's self-reported control over income and their ability to exercise true financial autonomy. Although 63% of women reported controlling their income, the significantly high number (94%) who reported limitations in gaining income autonomy suggests that their decision-making power over finances may be limited. This is also supported by the finding that 56% of women lack the access to or control over credit and financial services. Additionally, control over income suggests ability to solely or jointly decide on use of household income, while autonomy of income reflect one's motivation in use of income, i.e. do income decisions reflect personal choice or are motivated by external pressures. Therefore, we intriguingly observe that while 63% of the women report solely/jointly participating in all applicable income decisions, 94% report that their decisions were motivated by fear of other's disapproval rather than their own preference.

Additionally, most women lack the ability to visit important locations. According to Jennings et al. [27], the ability of a woman to visit important locations depends on whether they are permitted to do so by their spouse or older family members. Permission to visit is granted according to perceptions of their ability to fulfil household duties and observe *purdah* satisfactorily. Furthermore, almost half of the women report that they lack access to credit and financial services or are unable to make decisions about them. These inadequacies contribute strongly to the disempowerment of women among the respondents of our study in Bangladesh, as revealed by our findings.

Despite these findings, women have very high adequacies regarding input in decision making regarding aquaculture production. This phenomenon may be attributed to the sampling bias in the selection of women actively engaged in aquaculture for the pilot study, as they are likely to have a higher level of empowerment compared to the average Bangladeshi woman. In terms of ownership of other assets and land, our findings indicate very high adequacies.

Similarly, high adequacy scores were reported in achieving work balance, which is anomalous to other studies that suggest rural women almost always fare poorly in achieving work balance [28–30]. This finding is mainly due to the challenges during the data collection in the field where secondary time use for childcare was not recorded. The working hours of women are likely under-reported as childcare was not adequately captured. This could be another reason why the scores for 3DE, GPI and pro-WEFI are slightly higher than the scores for WEAI and pro-WEAI [25, 26].

Men achieve higher percentage of adequacies across most pro-WEFI indicators with the exception of membership in any groups and/or influential groups, work balance, and having respect among household members. According to our findings, these inadequacies are the major contributors to the disempowerment of men in the fish production sector in Bangladesh. The high inadequacies in group membership among men can be attributed to the fact that the existence of groups, especially influential ones, are very limited in the study area and most of the groups that exist are micro-credit groups where women dominate membership [25, 26, 31]. It is plausible that men fare relatively poorly in achieving work balance as work hours are more likely to be underestimated for women than men since childcare is not captured adequately, which in rural Bangladesh is disproportionately contributed by women [31–33]. It is important to also compare the censored and uncensored headcount ratios to understand the inadequacy of indicators in relation to the status of empowerment [6, 26]. For indicators where the uncensored and censored headcounts are very similar, it can be concluded that most, if not all, of the inadequacy comes from disempowered individuals. Such is the case for the censored and uncensored headcounts of inadequacy for women in achieving group membership and membership in influential groups, self-efficacy, access to credit, and restricted mobility, suggesting most women who are inadequate in the said indicators also fail to achieve empowerment overall.

Since each indicator in the pro-WEFI methodology is assigned an equal weight, it is important to evaluate the relationships among the indicators. Our findings show a weak correlation between all indicators, except for the correlation between group membership and membership in influential groups. This is because it is a necessary condition to be adequate in the former to be adequate in the latter. Malapit et al. [6] makes similar observations studying the associations between pro-WEAI indicators and explains that the choice to define the said indicators was deliberate to assign more weights to the collective agency which has very fewer indicators compared to the other two agencies. A high association between two indicators can imply more weights assigned to an indicator pair [6] and since all other indicators used to compose pro-WEFI are weakly correlated, each indicator, therefore, meaningfully contributes to the composite index with unique dimensions of agency that determine the empowerment of women in aquaculture. There is a high redundancy among all pro-WEFI indicators, as also observed by Malapit et al., [6] with pro-WEAI indicators. We mostly observe weak associations among indicators. Therefore, high redundancy does not pose major risk to the robustness of the weighted pro-WEFI index. However, high values of redundancy among indicators are suggestive that inadequacies are grouped.

From the comparison results of the 3DE scores and headcount ratio of empowerment for men and women considering computation of 3DE score using equal weights for all indicators, followed by pro-WEFI and computation of 3DE score using equal weights for three domains, we retain the computation of 3DE score using equal weights for all indicators, followed by pro-WEFI, as the chosen weighting scheme for pro-WEFI. This is because its observed ranking of men and women's scores and headcount ratio aligns with findings from other similar studies which show that men have higher empowerment scores than women in Bangladesh [10, 34]. Additionally, the number of indicators is not balanced across the three domains, which

would assign a heavier weight to the collective agency indicators compared to the instrumental and intrinsic agency indicators [6]. Because we have no theoretical basis for weighting some indicators higher than others, we have decided to give all indicators the same weight, which will further allow comparability across various projects targeted at respondents engaged in aquaculture.

## 5. Conclusion

In this study, we present the pro-WEFI methodology and results of a pilot study conducted to quantify women's empowerment and agency in aquaculture. To develop the pro-WEFI composite index, we used survey-based data from 217 households from several districts in Bangladesh participating in the IDEA project. The methods and results presented in this paper are based on a cognitively-tested survey instrument, through which men and women were asked to self-report the extent of their decision-making, agency and access to various resources, across three key domains. We find more than a third of the households surveyed did not achieve gender parity. Even though we find that the mean empowerment score of men to be higher than that of women, the mean 3DE score of women is higher than in other studies measuring women's empowerment in Bangladesh.

The decomposability of pro-WEFI into individual indicators further allowed us to identify where women face more challenges and areas of improvement to augment women's empowerment, specifically in the aquaculture sector. Women's lack of autonomy in the use of income, inability to visit important locations, lack of access to and decision-making about financial services, and inadequate self-efficacy are the most disempowering aspects for women in an aquaculture household. This study underscores the importance of the pro-WEFI and its domains and indicators that can help to: a) inform the development of targeted and effective interventions that build on the domains where gender inequality is most pronounced, and b) track changes in the adequacy of different indicators before and after an intervention and over time.

The study emphasizes the need to transform the gender norms that constrain women's economic participation (i.e., from equitable use of income and participating in markets), and decision-making. To achieve this, future interventions need to focus on addressing deeply embedded power imbalances and discriminatory norms through gender-transformative approaches (GTA). By addressing power imbalances and discriminatory norms through identifying the complex interplay of gender and other social factors, interventions can be tailored to the specific barriers faced by women face in key domains such as economic participation and decision-making, as identified in studies such as this one.

It is equally important to identify some of the key limitations of the study in order to understand what opportunities there are to advance research so that the index evolves to provide a more accurate diagnosis of gender inequality in the aquaculture sector. One key limitation in the application of the tool lies in the sample size and characteristics of the women surveyed. As the sample on which this study is based is small (i.e. 217 women), the results are difficult to generalize and also differ from other studies whose samples are representative of the study population. Another limitation of the results of the study lies in the challenges of collecting data in the field. In this study, there were difficulties with data collection, particularly in the self-assessment of complex and time-consuming modules, especially the module on time allocation.

The time allocation module was initially filled out separately on paper and later entered into a CAPI software to make it easier to answer. However, this module may have led to inaccurate time data, as secondary time use for childcare was not recorded. As a result, working hours may have been underreported, making it difficult to accurately represent the time spent

by women on reproductive activities. Careful consideration and implementation of the time use module is required to ensure that future studies using the pro-WEFI tool accurately capture women's time use.

Finally, this study was limited by the absence of a qualitative assessment of women's agency in the study location, which could not be included due to budgetary restrictions. The lack of in-depth qualitative assessment of women's agency in the three domains of empowerment in aquaculture hinders subjectivity and contextual understanding (i.e. contextual and individual experiences of gender and social norms), limits the ability to capture power dynamics within households and communities, and hinders understanding of other factors that contribute to empowerment/disempowerment.

## Supporting information

**S1 Datasets. This is the project level women's empowerment in fisheries and aquaculture index (pro-WEFI) in Bangladesh datasets.**
(XLSX)

**S1 Questionnaires. This is the quantitative questionnaire that was used for data collection in Bangladesh.**
(DOCX)

## Acknowledgments

We would like to thank the Resilient Aquatic Food Systems for Healthy People and Planet (RAqFS) of the One CGIAR research and development initiative for providing the platform to carry out this study. We would also like to thank the KIT Royal Tropical Institute and Katie Sproule for their collaboration in the development and validation of the Pro-WEFI tool. We also thank the Ecociate Consultants Pvt. Ltd for managing the field work for this study. Finally, we thank Colin Shelley (PhD) and Alvaro Paz Mendez for their leadership of the IDEA project and Cathy Rozel Farnworth (PhD) for reviewing the manuscript.

## Author Contributions

**Conceptualization:** Rahma I. Adam, Surendran Rajaratnam, Farha Deba Sufian.

**Data curation:** Lucy Njogu.

**Formal analysis:** Rahma I. Adam, Farha Deba Sufian, Lucy Njogu.

**Funding acquisition:** Rahma I. Adam, Surendran Rajaratnam.

**Investigation:** Rahma I. Adam.

**Methodology:** Rahma I. Adam, Surendran Rajaratnam, Farha Deba Sufian, Lucy Njogu.

**Project administration:** Rahma I. Adam, Surendran Rajaratnam.

**Resources:** Rahma I. Adam, Surendran Rajaratnam.

**Software:** Farha Deba Sufian.

**Supervision:** Rahma I. Adam, Surendran Rajaratnam.

**Validation:** Rahma I. Adam, Surendran Rajaratnam.

**Visualization:** Rahma I. Adam, Farha Deba Sufian, Lucy Njogu.

**Writing – original draft:** Rahma I. Adam, Surendran Rajaratnam, Farha Deba Sufian, Lucy Njogu.

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
