## [Decision Letter · Decision Letter 0]

19 Jul 2023

PONE-D-23-09595The state of women’s empowerment in the aquaculture sector: the case of BangladeshPLOS ONE

Dear Dr. Adam,

Thank you for submitting your manuscript to PLOS ONE. After careful consideration, we feel that it has merit but does not fully meet PLOS ONE’s publication criteria as it currently stands. Therefore, we invite you to submit a revised version of the manuscript that addresses the points raised during the review process.

Authors need to address the key issues indicated by both the reviewers:1. revision of abstract and introduction. 2. strengthening the methodology section.3. improving discussion. Comments of both the reviewers are attached. ==============================

We look forward to receiving your revised manuscript.

Kind regards,

Muhammad Khalid Bashir, PhD

Academic Editor

PLOS ONE

Journal Requirements:

5. Please ensure that you include a title page within your main document. You should list all authors and all affiliations as per our author instructions and clearly indicate the corresponding author.

Reviewers' comments:

Reviewer's Responses to Questions

**Comments to the Author**

1. Is the manuscript technically sound, and do the data support the conclusions?

Reviewer #1: No

Reviewer #2: No

2. Has the statistical analysis been performed appropriately and rigorously? 

Reviewer #1: Yes

Reviewer #2: No

3. Have the authors made all data underlying the findings in their manuscript fully available?

Reviewer #1: Yes

Reviewer #2: No

4. Is the manuscript presented in an intelligible fashion and written in standard English?

Reviewer #1: No

Reviewer #2: No

5. Review Comments to the Author

Reviewer #1: Apart from the first page and a half, the paper is well written and the methods and data analysis are fine. However, I have some serious concerns about the way the pro-WEFI was designed and used to measure empowerment. I have explained my concerns in the attached copy of the manuscript.

Reviewer #2: Thank you for giving me the opportunity to review this paper. The topic is important and interesting, but the manuscript needs the authors’ attention before acceptance. I have noted the following deficiencies that should be addressed in revision:

Authors should revise the whole abstract section by having the following unstructured sections: purpose, methodology, results, and conclusion.

"and" and "agency" should be removed from keywords.

Introduction needs additional work and should have the following things:

1) Research gap 2) Study objective 3) Research questions or hypotheses 4) Study contribution

References throughout the manuscript should be updated and use recent studies from 2018–2023, especially in the introduction section. The following studies might help in this regard:

https://doi.org/10.3390/agriculture12081161

https://doi.org/10.1016/j.worlddev.2020.105292

"Background and motivation for creating and implementing pro-WEFI" should be changed to review of literature and should include findings of the previous studies. Hypotheses should also be constructed on the basis of this literature review. Also, provide a graphical presentation of the conceptual framework of the study.

Line 161: Why was the survey questionnaire tested in India and not in Bangladesh? Surprising

Please add a section called "discussion" to discuss your results thoroughly. I am wondering if a paper can be written without discussing the results in light of previous literature.

The background statistics presented in Table 1 are not enough to show the background of the respondents.

The authors should come up with some solid results. Currently, the data has been presented in a vague manner. The authors should draw some conclusions based on their results.

Where are the limitations of the study?

Please also use some statistical models to improve the credibility of your results.

6. PLOS authors have the option to publish the peer review history of their article (what does this mean?). If published, this will include your full peer review and any attached files.

Reviewer #1: No

Reviewer #2: No

---

## [Author Response · Author response to Decision Letter 0]

18 Nov 2023

Response to reviewers’ comments

Reviewer 1 comments Response

I recommend that the abstract is revised. The abstract has been revised

This paper does not measure impact. I suspect that this paper presents the results of a baseline that was conducted as part of the IDEA project. I assume that this is why you collected data from three different groups, even though the paper does not present the data per group. I assume that you will make the comparisons when you have the mid or endline data. The study assessed the impact of the IDEA project on women’s empowerment.

The respondents interviewed as part of the pro-WEFI implementation study included men and women beneficiaries of the IDEA project’s nutrition, aquaculture, and gender training.

I suggest that you rewrite the abstract to better reflect what this paper is all about. The abstract has been re-written

In general, the introduction is not well written and should be significantly improved. It is markedly worse than the rest of the paper and seems to have been written by a different person The introduction has been revised

Word choice, this is the second time pivotal is used in this sentence. Not in the revised introduction

Why do you say "on the other hand" here? It does not make sense. Not in the revised introduction

What decision making are they excluded from? Aquaculture management? Value chain related decisions (what species to grow, when to harvest, when to sell)? Etc. Not in the revised introduction

Is it correct to say that the paper implemented the survey? I'd think the implementer was the IDEA project. Not in the revised introduction

Why would you collect names in the first place and why replace it with a pseudonym rather than a number? This has been corrected and the statement removed from the article

It is interesting that you collected data from the different groups of people. How many persons did you interview in each grouping?

Also, you explain here that you interviewed the three groups of people, but you don't present the data by the three groups. All the data is presented as one uniform group and you are not analyzing if there are differences between the three groups. What is the rationale for interviewing three different groups if you do not follow up by analyzing the data to compare the groups? The respondents interviewed as part of the pro-WEFI implementation study included men and women beneficiaries of the IDEA project’s nutrition, aquaculture, and gender training.

I feel that the authors miss an important point here. I assume that they did not expect to find such similar scores between men and women. I believe there are a couple of reasons for this finding

1. Both men and women living in rural Bangladesh societal, governance, and economic hardships. Struggling is not unique to women only.

2. The questions are identifying to what extent men and women feel empowered within the three domains, but it does not ask people to compare the status of women to that of men. It is only asking people to give a perception of their own situation.

3. The questions are getting at to what extent people feel empowered. However, they do not get at power imbalances between men and women, social injustices, and cultural differences between for example how girls and boys are raised. The fact that women are not allowed to go to the market without permission from their husbands is an indication of large power imbalances and social injustice between women and men. The description of the results has been revised and now includes “Overall, men have fewer inadequacies than women, whose distribution lie slightly to the right of men’s distribution of inadequacies in the shaded area. However, the intensity of disempowerment experienced by men and women in the sample are very comparable, as both distributions in the figure follow a very similar pattern for 4 or more inadequacies. This observation is line with the results exhibited in the high-level indices summary in Table 2, where both headcount of respondents not achieving empowerment and the average disempowerment score for women are nearly the same as that of men, suggesting similar intensity of disempowerment.”

I find this impediment problematic because you don't explain the root causes to this until the conclusions. There is so much to unpack in the inability to visit important locations. The fact that women are not empowered to make their own decisions about when and where to go, indicates a deeply embedded lack of empowerment. However, in the paper, it is treated as a shallow factor, almost like a physical inability. The root causes have been added in the discussion section.

You could do a lot more with the conclusions. While you talk about the areas where women are disempowered, what are the strengths in women's empowerment that future aquaculture projects. Also, what are the weaknesses of the tool and what would you do differently - or what follow on studies would you propose to get deeper into this topic? The weaknesses of the tool have been discussed under the limitations of the study.

I don't think this paper leads us to this conclusion. The survey did not give us any information about women's contribution to the aquaculture sector. Furthermore, women in general seemed to score higher than men in "input in productive decisions" financial services decisions" and "control over use of income", while they scored poorly on women's ability to visit important locations. I think this is a case where your data does not line up well with what you thought you would find. The findings changed after the data analysis was redone. Lack of autonomy in the use of income, was the most disempowering aspect for women.

The conclusions have been revised.

Reviewer 2 comments Response

Authors should revise the whole abstract section by having the following unstructured sections: purpose, methodology, results, and conclusion.

 "and" and "agency" should be removed from keywords.

 The abstract has been revised

Introduction needs additional work and should have the following things:

1) Research gap 2) Study objective 3) Research questions or hypotheses 4) Study contribution

 The introduction has been revised

References throughout the manuscript should be updated and use recent studies from 2018–2023, especially in the introduction section. The following studies might help in this regard:

https://doi.org/10.3390/agriculture12081161

https://doi.org/10.1016/j.worlddev.2020.105292

Some of the older references e.g. Rowlands, 1995 give important backgrounds on the topic.

"Background and motivation for creating and implementing pro-WEFI" should be changed to review of literature and should include findings of the previous studies The sub-section "Background and motivation for creating and implementing pro-WEFI" has been removed

Hypotheses should also be constructed on the basis of this literature review. The introduction has been revised to include the hypotheses

Also, provide a graphical presentation of the conceptual framework of the study. Figure 1 comprehensively shows the domains and indicators of Pro-WEFI (Source: McDougall, et al. 2022)

Line 161: Why was the survey questionnaire tested in India and not in Bangladesh? Surprising Since the study was undertaken during covid-19 period with movement restriction , the survey was tested in India as the communities share the same characteristics and agro ecological zones

Please add a section called "discussion" to discuss your results thoroughly. I am wondering if a paper can be written without discussing the results in light of previous literature The discussion section has been added

The background statistics presented in Table 1 are not enough to show the background of the respondents.

 The table has been revised after data reanalysis

The authors should come up with some solid results. Currently, the data has been presented in a vague manner. The authors should draw some conclusions based on their results.

 The results, discussion and conclusions sections have been revised

Where are the limitations of the study?

 The limitations of the study has been included

Please also use some statistical models to improve the credibility of your results.

 We use Cramer’s V to identify the strength of the correlation between pro-WEFI indicators.

Reviewer 2 comments Response

Authors should revise the whole abstract section by having the following unstructured sections: purpose, methodology, results, and conclusion.

 "and" and "agency" should be removed from keywords.

 The abstract has been revised

Introduction needs additional work and should have the following things:

1) Research gap 2) Study objective 3) Research questions or hypotheses 4) Study contribution

 The introduction has been revised

References throughout the manuscript should be updated and use recent studies from 2018–2023, especially in the introduction section. The following studies might help in this regard:

https://doi.org/10.3390/agriculture12081161

https://doi.org/10.1016/j.worlddev.2020.105292

Some of the older references e.g. Rowlands, 1995 give important backgrounds on the topic.

"Background and motivation for creating and implementing pro-WEFI" should be changed to review of literature and should include findings of the previous studies The sub-section "Background and motivation for creating and implementing pro-WEFI" has been removed

Hypotheses should also be constructed on the basis of this literature review. The introduction has been revised to include the hypotheses

Also, provide a graphical presentation of the conceptual framework of the study. Figure 1 comprehensively shows the domains and indicators of Pro-WEFI (Source: McDougall, et al. 2022)

Line 161: Why was the survey questionnaire tested in India and not in Bangladesh? Surprising Since the study was undertaken during covid-19 period with movement restriction , the survey was tested in India as the communities share the same characteristics and agro ecological zones

Please add a section called "discussion" to discuss your results thoroughly. I am wondering if a paper can be written without discussing the results in light of previous literature The discussion section has been added

The background statistics presented in Table 1 are not enough to show the background of the respondents.

 The table has been revised after data reanalysis

The authors should come up with some solid results. Currently, the data has been presented in a vague manner. The authors should draw some conclusions based on their results.

 The results, discussion and conclusions sections have been revised

Where are the limitations of the study?

 The limitations of the study has been included

Please also use some statistical models to improve the credibility of your results.

 We use Cramer’s V to identify the strength of the correlation between pro-WEFI indicators.

---

## [Decision Letter · Decision Letter 1]

12 Jan 2024

PONE-D-23-09595R1Development of Project level Women's Empowerment in Fisheries and Aquaculture Index (Pro-WEFI): A Bangladesh Case StudyPLOS ONE

Dear Dr. Adam,

Thank you for submitting your manuscript to PLOS ONE. After careful consideration, we feel that it has merit but does not fully meet PLOS ONE’s publication criteria as it currently stands. Therefore, we invite you to submit a revised version of the manuscript that addresses the points raised during the review process.

**Dear authors as both the reviewers are concerned about the data and one of them want to have a look at your data set, your are requested to deal with this issue carefully. ** Please submit your revised manuscript by Feb 26 2024 11:59PM. If you will need more time than this to complete your revisions, please reply to this message or contact the journal office at plosone@plos.org. Please include the following items when submitting your revised manuscript:A rebuttal letter that responds to each point raised by the academic editor and reviewer(s). You should upload this letter as a separate file labeled 'Response to Reviewers'.A marked-up copy of your manuscript that highlights changes made to the original version. You should upload this as a separate file labeled 'Revised Manuscript with Track Changes'.An unmarked version of your revised paper without tracked changes. You should upload this as a separate file labeled 'Manuscript'.

We look forward to receiving your revised manuscript.

Kind regards,

Muhammad Khalid Bashir, PhD

Academic Editor

PLOS ONE

Reviewers' comments:

Reviewer's Responses to Questions

**Comments to the Author**

1. If the authors have adequately addressed your comments raised in a previous round of review and you feel that this manuscript is now acceptable for publication, you may indicate that here to bypass the “Comments to the Author” section, enter your conflict of interest statement in the “Confidential to Editor” section, and submit your "Accept" recommendation.

Reviewer #2: (No Response)

2. Is the manuscript technically sound, and do the data support the conclusions?

Reviewer #2: Partly

3. Has the statistical analysis been performed appropriately and rigorously? 

Reviewer #2: I Don't Know

4. Have the authors made all data underlying the findings in their manuscript fully available?

Reviewer #2: No

5. Is the manuscript presented in an intelligible fashion and written in standard English?

Reviewer #2: No

6. Review Comments to the Author

Reviewer #2: Dear authors

I want to have a look at your data before further reviewing this paper. If you look at journal guidelines, authors needs to provide data on with the manuscript submission in this journal. So, I could not find any data in attached files. It is not enough to say "Data is available upon request" that you mentioned in data availability statement.

7. PLOS authors have the option to publish the peer review history of their article (what does this mean?). If published, this will include your full peer review and any attached files.

Reviewer #2: No

---

## [Author Response · Author response to Decision Letter 1]

20 May 2024

Reviewers # 1 and #2 responses: (1) We have worked hard and well to make sure that the findings of the research are explained using detailed language, so that it is easier to understand for a wider audience and we have explained the methodology and the data presented to greater depths. For instance, see the attached revised manuscript lines 416-428 and lines 80-90. (2) We had already shared the data in our first submission, it is just that statement was missing in the paper that states that data is available and we have shared as supplementary material, we have now included that statement in the paper, please see line 561, but actually was available in the online system, it was by mistake that we noted that data is available upon request, because a lot of manuscripts tend to be published like that. (3) We have worked with a native speaker of English from our organization, who is also a scientist to make sure that the language is smooth and no any grammatical errors.

---

## [Decision Letter · Decision Letter 2]

24 Jul 2024

PONE-D-23-09595R2Development of Project level Women's Empowerment in Fisheries and Aquaculture Index (Pro-WEFI): A Bangladesh Case StudyPLOS ONE

Dear Dr. Adam,

Thank you for submitting your manuscript to PLOS ONE. After careful consideration, we feel that it has merit but does not fully meet PLOS ONE’s publication criteria as it currently stands. Therefore, we invite you to submit a revised version of the manuscript that addresses the points raised during the review process.

Authors need to pay attention to the reviewer 3's comments and suggestions very carefully and revise the paper accordingly. Authors are also requested to change the title after making the necessary changes. 

We look forward to receiving your revised manuscript.

Kind regards,

Muhammad Khalid Bashir, PhD

Academic Editor

PLOS ONE

Reviewers' comments:

Reviewer's Responses to Questions

**Comments to the Author**

1. If the authors have adequately addressed your comments raised in a previous round of review and you feel that this manuscript is now acceptable for publication, you may indicate that here to bypass the “Comments to the Author” section, enter your conflict of interest statement in the “Confidential to Editor” section, and submit your "Accept" recommendation.

Reviewer #2: All comments have been addressed

Reviewer #3: (No Response)

2. Is the manuscript technically sound, and do the data support the conclusions?

Reviewer #2: Yes

Reviewer #3: Partly

3. Has the statistical analysis been performed appropriately and rigorously? 

Reviewer #2: Yes

Reviewer #3: Yes

4. Have the authors made all data underlying the findings in their manuscript fully available?

Reviewer #2: Yes

Reviewer #3: Yes

5. Is the manuscript presented in an intelligible fashion and written in standard English?

Reviewer #2: Yes

Reviewer #3: Yes

6. Review Comments to the Author

Reviewer #2: Than you for addressing my comments. I suggest accepting this manuscript in current form..........................

Reviewer #3: Manuscript ID: PONE-D-23-09595R2

The title is unclear. What do you mean by the development of Pro-WEFI? If you mean that you have developed Pro-WEFI for Bangladesh for the first time, it has already been done by the following paper: “DOI: 10.1177/21582440241250114,” published in 2024 in Sage Open. It is the first published paper on Pro-WEFI in Bangladesh. Contrarily, if you mean you have estimated the women empowerment index using Pro-WEFI for aquaculture in the northern part of Bangladesh, then the title may fit with revision.

As mentioned by the first reviewer in the first round of revision: “This paper does not measure impact. I suspect that this paper presents the results of a baseline that was conducted as part of the IDEA project. I assume that this is why you collected data from three different groups, even though the paper does not present the data per group. I assume that you will make the comparisons when you have the mid or endline data.” This study focuses only on the status of women empowerment in the northwestern part of Bangladesh, specifically those involved in aquaculture. Therefore, the title could be "Women Empowerment Status" or "Status Quo of Women Empowerment: A Case of the Aquaculture Community in the Northwestern Part of Bangladesh." If the authors insist on the current title, it adds nothing new for the reader.

The term 'fisheries' is broader, including capture, culture, and marine fisheries in Bangladesh. The previous Pro-WEFI paper focused on marine capture fishing communities. This paper focuses on culture fish farmers, not fisheries and aquaculture both.

The sample size is only 217, which is a bit low to accurately determine the women empowerment index. Proper justification of the sampling frame is needed. You may review previous literature (see the above paper and Table 2a).

Line 133: Bogura, Naogaon, and Rangpur are districts, not divisions.

Line 134: In the abstract section, the number of households (HH) is 227, but in the methodology section, it is 217.

Line 701: In Table 2a, compare the results with this paper (DOI: 10.1177/21582440241250114).

Section 2.1.2: The number of male respondent HH is 211, but in Table 2, it is 208.

Line 20: All HH surveyed were dual adult HH, but dual adult HH would be either 208 or 211.

Table 1: Last row, “dual adult HH,” is inconsistent. Table 2 shows dual adult HH would be either 208 or 211 out of 217 or 227.

Table 9: Why does "equally weighted by domain" show women have more empowerment than men?

Figure 2: The fourth indicator, “Respect among household members,” shows more inadequacy, whereas Table 3 shows the highest inadequacy score is for autonomy in income.

Pages 46 and 47 (in R2 version): Why do these three figures appear again?

Table 3 Headcount Ratios: The last indicator, “membership in influential groups,” shows a higher inadequacy ratio for men. Why is it higher for men? It should be lower for men.

7. PLOS authors have the option to publish the peer review history of their article (what does this mean?). If published, this will include your full peer review and any attached files.

Reviewer #2: No

Reviewer #3: **Yes: **Md. Salauddin Palash

---

## [Author Response · Author response to Decision Letter 2]

19 Oct 2024

Response to reviewers 

Comment #1. This paper does not measure impact. I suspect that this paper presents the results of a baseline that was conducted as part of the IDEA project. I assume that this is why you collected data from three different groups, even though the paper does not present the data per group. I assume that you will make the comparisons when you have the mid or endline data.” This study focuses only on the status of women empowerment in the northwestern part of Bangladesh, specifically those involved in aquaculture. Therefore, the title could be "Women Empowerment Status" or "Status Quo of Women Empowerment: A Case of the Aquaculture Community in the Northwestern Part of Bangladesh." If the authors insist on the current title, it adds nothing new for the reader.

Response to comment #1: We have now changed the title of the paper as you have suggested for us, and indeed the tool has already been used by other researchers, though it had not been cognitively tested systematically. And indeed, what we have done in this study is to measure the status of women’s empowerment in North-western Bangladesh. Thus, the title is now, Measuring Women’s Empowerment in Aquaculture in Northwestern Bangladesh Using a Project level Women's Empowerment in Fisheries Index (Pro-WEFI).

Comment #2: The term 'fisheries' is broader, including capture, culture, and marine fisheries in Bangladesh. The previous Pro-WEFI paper focused on marine capture fishing communities. This paper focuses on culture fish farmers, not fisheries and aquaculture both. 

Response to comment #2: Yes indeed, we have removed the term fisheries, where it was inappropriately used and we have called it aquaculture. 

Comment #3.The sample size is only 217, which is a bit low to accurately determine the women empowerment index. Proper justification of the sampling frame is needed. You may review previous literature (see the above paper and Table 2a). 

Response to comment #3: Small sample size is mentioned as a limitation in the discussion section from line 550-554; Section 2.1.2 which discusses sample would elaborate on why only 217 households were surveyed. The reason for the sample size to be low is because the research budget was low, we did not have resources to expand the sample size. 

Comment #4. Line 134: In the abstract section, the number of households (HH) is 227, but in the methodology section, it is 217. 

Response to comment #4: 227 is a mistake in the abstract; it should be 217. Action performed: We have made the changes in the revised attached manuscript as required. 

Comment #5: Line 701: In Table 2a, compare the results with this paper (DOI: 10.1177/21582440241250114). 

Response to comment #5: Table 2a has been revised by including results from the manuscript suggested by the reviewers and corresponding changes have been made to the draft.

Comment #6: Section 2.1.2: The number of male respondents HH is 211, but in Table 2, it is 208. 

Response to comment #6: The survey has 217 female and 208 male responses from a total of 217 households; 211 is a mistake. 

We have revised it in line 153.

Comment #7. Line 20: All HH surveyed were dual adult HH, but dual adult HH would be either 208 or 211. 

Response to comment #7: All 217 households surveyed are dual-adult households; however, of the 217 households, 208 complete male surveys were done. So, the number of dual-responses available to calculate Gender Parity Index was from 208 households (as outlined in table 2). 

We have revised line items to “Number of Dual-response households” in Table 2. 

Comment # 8:Table 1: Last row, “dual adult HH,” is inconsistent. Table 2 shows dual adult HH would be either 208 or 211 out of 217 or 227. 

Response to comment #8: Table 1 shows that 100% of the household surveyed were Dual-Adult households, suggesting the households had at least one adult male and one adult female member. This means All 217 households surveyed are dual-adult households; however, of the 217 households, 208 complete male surveys were done; So, the number of dual-responses available to calculate Gender Parity Index was from 208 households (as outlined in table 2). 

Comment #9: Table 9: Why does "equally weighted by domain" show women have more empowerment than men? 

Response to comment #9: Already discussed in details in line 363-377 (page 16).

Comment #10: Figure 2: The fourth indicator, “Respect among household members,” shows more inadequacy, whereas Table 3 shows the highest inadequacy score is for autonomy in income. 

Response to comment #10: I think Figure 3 is being referred to (not figure 2). However, in figure 3 the bar for autonomy is higher in length than Respect among household members (consistent with censored headcount results in table 3).

Comment #11: Pages 46 and 47 (in R2 version): Why do these three figures appear again? 

Response to comment #11: We have removed them. 

Comment #12: Table 3 Headcount Ratios: The last indicator, “membership in influential groups,” shows a higher inadequacy ratio for men. Why is it higher for men? It should be lower for men. 

Response to comment #12: In our sample we find adequacy in both "group membership" and “membership in influential groups" to be slightly higher for women than men. The reason for this is explained in the manuscript in the discussion section, line 456-459: "The high inadequacies in group membership among men can be attributed to the fact that the existence of groups, especially influential ones, are very limited in the study area and most of the groups that exist are micro-credit groups where women dominate membership (Sraboni et al., 2014; Akhter & Cheng, 2020; Waid et al., 2022). "

---

## [Decision Letter · Decision Letter 3]

8 Nov 2024

Measuring Women’s Empowerment in Aquaculture in Northwestern Bangladesh Using a Project level Women's Empowerment in Fisheries Index (Pro-WEFI)

PONE-D-23-09595R3

Dear Dr. Adam,

We’re pleased to inform you that your manuscript has been judged scientifically suitable for publication and will be formally accepted for publication once it meets all outstanding technical requirements.

Kind regards,

Muhammad Khalid Bashir, PhD

Academic Editor

PLOS ONE

Additional Editor Comments (optional):

Reviewers' comments:

Reviewer's Responses to Questions

**Comments to the Author**

1. If the authors have adequately addressed your comments raised in a previous round of review and you feel that this manuscript is now acceptable for publication, you may indicate that here to bypass the “Comments to the Author” section, enter your conflict of interest statement in the “Confidential to Editor” section, and submit your "Accept" recommendation.

Reviewer #2: All comments have been addressed

Reviewer #3: All comments have been addressed

2. Is the manuscript technically sound, and do the data support the conclusions?

Reviewer #2: Yes

Reviewer #3: Yes

3. Has the statistical analysis been performed appropriately and rigorously? 

Reviewer #2: Yes

Reviewer #3: Yes

4. Have the authors made all data underlying the findings in their manuscript fully available?

Reviewer #2: Yes

Reviewer #3: Yes

5. Is the manuscript presented in an intelligible fashion and written in standard English?

Reviewer #2: Yes

Reviewer #3: Yes

6. Review Comments to the Author

Reviewer #2: Thank you for revising the manuscript according to my suggestions. Best of luck ................................

Reviewer #3: Thanks for revising the manuscript. Authors address all the comments as much as possible. It may now proceed for publication.

7. PLOS authors have the option to publish the peer review history of their article (what does this mean?). If published, this will include your full peer review and any attached files.

Reviewer #2: No

Reviewer #3: **Yes: **Md Salauddin Palash

---

## [Editor Report · Acceptance letter]

26 Dec 2024

PONE-D-23-09595R3 

PLOS ONE

Dear Dr. Adam, 

I'm pleased to inform you that your manuscript has been deemed suitable for publication in PLOS ONE. Congratulations! Your manuscript is now being handed over to our production team.

Kind regards, 

on behalf of

Dr. Muhammad Khalid Bashir 

Academic Editor

PLOS ONE